

# Subsurface floats in the Filchner Trough provide first direct under-ice tracks of eddies and circulation on shelf

Jean-Baptiste Sallée[1], Lucie Vignes[1], Audrey Minière[2], Nadine Steiger[1], Etienne Pauthenet[1,5], Antonio Lourenco[1], Kevin Speer[3], Peter Lazarevich[3], and Keith W Nicholls[4]

[1]Laboratoire d'Océanographie et du Climat, Sorbonne Université / CNRS, Paris, France
[2]Mercator Ocean International, Université Toulouse III - Paul Sbatier, Toulouse, France
[3]Geophysical Fluid Dynamics Institute and Department of Scientific Computing, Florida State University, Tallahassee, USA
[4]British Antarctic Survey, Cambridge, UK
[5]Univ Brest, Ifremer, CNRS, IRD, LOPS, F-29280 Plouzané, France

**Correspondence:** Jean-Baptiste Sallée (jean-baptiste.sallee@locean.ipsl.fr)

**Abstract.**

Bottom water formation in the Weddell Sea and mass loss from the Filchner-Ronne Ice Shelf are tightly linked by the supply of Warm Deep Water to the continental shelf. Heavy sea ice cover and icebergs restrict ship access and upper ocean measurements by moorings, compelling us to try new sampling methods. We present results from the first dedicated under-

sea-ice float experiment tracking circulation on the continental shelf between the Brunt Ice Shelf and Filchner Ice Shelf. Seven Apex profiling floats were deployed in 2017 at three different locations, targeting the sources of modified Warm Deep Water (mWDW) inflow and Ice Shelf Water (ISW) circulation in the Filchner Trough. The floats capture a warm mWDW regime with southward inflow over the eastern continental shelf and a cold ISW regime with a recirculation of ISW in the Filchner Trough throughout the four years of Pobservations. The mWDW flowing onto the continental shelf follows two pathways: the eastern

flank of the Filchner Trough and via a Small Trough on the shallow shelf farther east. In the present circulation regime, this warm water is blocked from reaching the ice shelf cavity due to the presence of the thick ISW layer inside the Filchner Trough. The floats' trajectories and hydrography reveal the dynamically active front, flow reversal, and eddy generation between these two water masses along the eastern flank of the Filchner Trough.

## 1   Introduction

The ocean circulation at the Antarctic margin controls the amount of heat that can be transported toward ice shelves surrounding Antarctica (Thompson et al., 2018). Ice shelves are the floating extensions of the Antarctic Ice Sheet and help stabilize the ice sheet's flow toward the ocean (Dupont and Alley, 2005; Reese et al., 2018). By modulating ocean heat reaching ice shelf cavities, the ocean circulation over the Antarctic continental shelves has a direct impact on the Antarctic Ice Sheet's mass balance and therefore global sea-level rise (Joughin et al., 2012; DeConto and Pollard, 2016). In turn, modulation in the

freshwater input to the ocean through ocean-driven basal melting of the ice shelves affects water mass formation and ocean circulation, and in particular dense water production feeding the global overturning circulation (Lago and England, 2019;





Moorman et al., 2020). Understanding the ocean circulation over the Antarctic continental shelf, and how it varies, therefore represents a key endeavour in our understanding of global climate.

On the Antarctic continent, the southern Weddell Sea is a key region for the formation of Antarctic Bottom Water that fills
the bottom of the world oceans (Foldvik and Gammelsrød, 1988; Meredith et al., 2013). This region hosts the Filchner-Ronne Ice Shelf (FRIS), Antarctica's largest ice shelf in volume (Fox et al., 1994). Its underlying cavity is categorized as a "cold and dense" cavity, associated with relatively low basal melt rates and dense water masses (Nicholls et al., 2009). This ice shelf regime is distinguished from "warm cavities" characterized by relatively large basal melt rates (Paolo et al., 2015; Rignot et al., 2019) and "cold and fresh cavities" associated with relatively low basal melt and light water masses (Hattermann et al., 2014;
Silvano et al., 2016). These different regimes are maintained by ocean-cryosphere feedback involving the ice shelves, sea ice, and ocean circulation (Jacobs et al., 1992; Thompson et al., 2018).

The FRIS in the Weddell Sea has had a relatively stable mass balance over past decades compared to warm cavity ice shelves west of the Antarctic Peninsula (Rignot et al., 2019); it is thought to be relatively protected from the warming influence of the Circumpolar Deep Water (CDW) of the Southern Ocean (Schmidtko et al., 2014) as a result of the presence of the Weddell
Gyre system (Vernet et al., 2019), but also because of the local frontal dynamics on the continental slope associated with the cold and dense water over the shelf (Thompson et al., 2018). The circulation and hydrography on the shelf is dominated by the production of High Salinity Shelf Water (HSSW, with temperatures at the surface freezing point $\sim$-1.9$^o$C) that is formed by sea ice formation on the shelf, specifically within coastal polynyas (Haid and Timmermann, 2013). This dense water mass enters the ice shelf cavity at its western entrance and transforms to a slightly colder and fresher Ice Shelf Water (ISW; temperatures
<-1.9$^o$C) through interaction with the ice shelf while it circulates inside the cavity. ISW exits the cavity through the Filchner Trough that forms a relatively shallow channel across the eastern continental shelf from the shelf break towards the ice shelf (Nicholls et al., 2009). Below about 500 m the trough is a semi-closed basin retaining ISW behind its sill. While observations have shown that the ISW flows northward along the eastern flank of the Filchner Trough (Darelius et al., 2014; Ryan et al., 2017), there is also evidence that the ISW seasonally exits the cavity over the western flank (Darelius and Sallée, 2018). Once
the ISW overflows the continental shelf across the sill of the Filchner Trough (Darelius et al., 2009; Foldvik et al., 2004) it contributes to the formation of Weddell Sea Deep Water and Weddell Sea Bottom Water, the precursors of Antarctic Bottom Water. Recent studies have shown a reduction in volume of the Weddell Sea Bottom Water linked to changes in dense water production on the continental shelf (Zhou et al., 2023). While past work has revealed the processes involved in the formation and modification of ISW, detailed information about its pathways and residence times are still lacking.
The existence of a pool of dense water over the southern Weddell Sea continental shelf is thought to create a barrier for the advection of relatively warm water off the continental shelf toward the FRIS cavity. The warm water originates from the CDW that upwells in the Southern Ocean; in the Weddell Sea, it is entrained into the Weddell Gyre system and transformed into Warm Deep Water (WDW), a fresher and cooler version of CDW (Ryan et al., 2016). There are two pathways for WDW to enter the continental shelf and subsequently flow southward toward the ice shelf as a yet cooler and fresher version of the
WDW, referred to as modified WDW (mWDW): an eastern pathway, along the eastern flank of the Filchner Trough, and a western pathway, in the Central Trough west of the Berkner Bank (Fig. 1; Nicholls et al., 2009, 2008; Ryan et al., 2017). The





western pathway is associated with the largest heat content (Nicholls et al., 2008; Vignes et al.), and about half of the heat at the shelf break that actually reaches the FRIS edge (Davis et al., 2022). Most of the heat advected southward onto the continental shelf through the Filchner Trough is thought to recirculate northward or to be mixed away before reaching the FRIS cavity

(Dinniman et al., 2011; Darelius et al., 2016). The flow of mWDW into the Filchner Trough occurs seasonally during austral summer and is linked to the wind variability (Årthun et al., 2012; Daae et al., 2017; Ryan et al., 2017). In 2017 the mWDW layer over the eastern flank of the Filchner Trough was warmer than in previous years and persisted for a longer time period (Ryan et al., 2020), but it is unclear how far south the warm water reached. Thus far, mWDW has only been observed at the ice front near 79°S in 2013 (Darelius et al., 2016). However, the region has been suggested to be prone to a potential self-

reinforcing change in the ocean circulation that could abruptly shift its cavity regime, turning it from a cold and dense cavity into a warm cavity with important global consequences for deep water formation and sea level rise (Hellmer et al., 2012, 2017; Hazel and Stewart, 2020). As a result, efforts to understand the controls on the flux of warm water and heat transport rate to the FRIS cavity have received much attention in recent years (Årthun et al., 2012; Darelius et al., 2016; Ryan et al., 2017; Daae et al., 2017; Ryan et al., 2020; Daae et al., 2020).

Over recent decades, the flow of warm water onto the continental shelf has been sampled using ship-based observations (Conductivity-Temperature-Depth (CTD) casts), CTD-instrumented marine mammals, and mooring observations. Ship-based observations are only possible during summer and are mostly concentrated in regions lightly covered by sea-ice during this period. Marine mammal observations depend on seal behaviour (Labrousse et al., 2021), so cannot be targeted toward specific currents or water-masses; also, they do not usually cover the entire water column and have larger measurement and positioning

uncertainties than ship-based observations (Siegelman et al., 2019). Mooring observations over the continental shelf (e.g. Ryan et al., 2017; Darelius et al., 2014) sample at a fixed point throughout the year, but possible damage by drifting icebergs leaves the upper part of the water column vulnerable and rarely instrumented. Autonomous instruments such as Argo floats have not been deployed on the southern Weddell Sea continental shelf. The various strategies used to observe the ocean over the continental shelf complement each other, but important gaps can be identified in the present observational system in this

challenging ice-covered part of the ocean: a dearth of wintertime observations, a lack of observations of water-mass mixing, and the lack of reliable information on circulation routes and time scales.

In this study, we utilise an innovative approach for sampling water mass characteristics and circulation on the Weddell Sea continental shelf with the aim of partly addressing this observational gap. We present and use, for the first time in this region, observations from seven Lagrangian profiling floats that were deployed on the continental shelf in February 2017. Three

RAFOS sound sources were deployed specifically for this experiment, with RAFOS hydrophones on the floats being used to provide their location every six hours. Although the ocean environmental conditions of the Weddell Sea continental shelf make float deployments very risky, due to the large area covered by multi-year ice and the presence of ice shelf cavities where instruments can get lost, all of the floats survived at least one year and up to four years, providing a unique set of observations. A similar approach was applied to determine float tracks under the Dotson Ice Shelf in the Amundsen Sea (Girton et al., 2019)

at distances up to about 100km.



Our observations on the eastern Weddell Sea shelf allow us to describe the circulation in the Filchner Trough region and the water mass pathways based on the float trajectories and the hydrography profiles, some of which are taken in previously unexplored areas that are typically covered with sea ice. We are able to examine the Lagrangian pathways of the mWDW toward the ice shelf, its interaction with the ISW, and the residence time of the ISW within the Filchner Trough. Selected mooring observations and ship-based measurements from the same time period complement the float data are presented, and finally we discuss our findings in the context of previous knowledge from the area.

## 2   Data and Methods

### 2.1   Autonomous RAFOS-enabled profiling floats

In February 2017, seven autonomous profiling Apex floats were deployed from the R/V *James Clark Ross* (cruise number JR16004; Sallee, 2018) at three different locations on the Weddell Sea continental shelf (Fig. 1): three floats on the continental slope upstream of the Filchner Trough (∼26°1 W, 74°3 S); two at the eastern part of the Filchner Trough sill (∼30°4 W, 74°5 S); and two farther south in the deeper part of the Filchner Trough (∼32°4 W, 76°3 S). Each of the floats carried a SeaBird SBE41 sensor to measure conductivity (C), temperature (T), and pressure (P), as well as a hydrophone to record the timed sound source signals (Table 1). The RAFOS protocol (Rossby et al., 1986) uses the timed arrivals of signals from the sound sources to determine the position. All floats together provided in total 1106 CTD-profiles over a time span of up to four years, across several hundreds of kilometers between the Filchner and Brunt ice shelves. Six of these seven floats sampled 244 profiles located over the continental shelf between February 2017 and April 2018; one of the floats (P12681; Fig. 3) circulated northward after being deployed and did not sample the shelf regime and is not discussed further.

Profiles of absolute salinity (SA), conservative temperature (Θ), potential density referenced to the surface ($\sigma_0$), and pressure (P) were derived from the observed profiles using the revised seawater equation of state (McDougall and Barker, 2011). The profiling frequency was adjusted between summer and winter months: the floats profiled once per day in summer, and then once every five days the rest of the year. The floats are programmed to start their (ascending) profile from 1800 dbar, which is generally deeper than the continental shelf. This means that in most cases they descended to the sea floor before starting the profile. Between profiles, the floats drifted at a programmed parking depth, and took CTD measurements at this depth every six hours. The parking depth chosen depended on the deployment positions of the floats: the two southernmost floats, deployed inside the Filchner Trough were programmed to drift at 250 dbar to reduce the risk of drifting inside ice shelf cavities; the other floats were programmed to drift at 400 dbar to target typical depths where mWDW is found on the Weddell Sea continental shelf (Årthun et al., 2012; Nicholls et al., 2009; Foldvik et al., 1985).

When a float surfaced, its positioning data and other recorded measurements were telemetered via the Iridium satellite network. If a float were to get to the surface when sea ice was present, its sensors and antennas would likely be damaged. To reduce this risk, each float implemented a "sea ice avoidance" logic to predict the presence of sea ice from the observed ocean characteristics measured during ascent (Klatt et al., 2007; Wong and Riser, 2011; Porter et al., 2019; Silvano et al., 2019). If



the measured in situ temperature, averaged between 20–40 m, was less than -1.69°C, the float halted its ascent, descended to its parking depth and stored its data to transmit them at the next successful surfacing.

## 2.2 Positioning of the floats

The Filchner Trough is a region covered by sea ice during most of the year and by multi-year ice in some parts, preventing floats from surfacing and acquiring GPS positions for their profiles. In order to locate the hydrographic profiles under ice and to obtain high-resolution Lagrangian trajectories of the floats, three sound sources were moored in the eastern continental shelf (74.02° S, 28.07° W; 74.85° S, 30.38° W; 75.39° S, 28.64° W; Fig. 1). Each source transmitted a frequency modulated signal four times a day with a carrier of about 260 Hz, and frequency increasing linearly by 2.5 Hz during 80s seconds (Rossby et al., 1986). The floats were programmed with listening windows consistent with the sound source emission times, and saved the six highest correlations between transmitted and expected sound source signal, as well as the six times of arrival (TOA) of the sound signal corresponding to each of the six correlations. The stored set of six TOA and maximum correlations were then transmitted via the Iridium satellite network when the float eventually surfaced and used to triangulate the position of the floats during their drift.

The TOA are corrected for three main biases. First, the offset between the start of the float listening window and the time of sound emission for each source is removed from the received TOA; for each listening window, the float started listening one minute before the first source emission, 21 minutes before the second source, and 41 minutes before the third source emission. Second, the TOA were corrected for the linear drift of the float internal clock. Third, TOA were adjusted to account for the delay in activating the memory card that records them. The card's activation is delayed by a margin of 10 to 24 seconds after the initiation of the float's listening window, introducing a temporal offset within the TOA measurements. Each TOA was then quality checked using the corresponding maximum correlation, re-confirmed by eye, and outliers and unrealistic records removed, to eventually form a clean and validated time-series of TOA for each float. The time-series of TOA was then smoothed with a loess filter removing frequencies higher than once per day to produce a better estimate of the daily mean position used here.

The corrected TOA were converted into distances between the float and the sound source using a fixed sound speed in water of 1455 m s$^{-1}$, derived from CTD measurements. Given that the float's hydrophones sampled every 0.3075 s during their listening window, the minimum achievable spatial resolution of the triangulation was $\Delta d$ = 477 m. Distances from the three sources were then converted into geographical position by minimizing the error of the triangulation procedure. The geographical position was computed for all time steps in which three sound source signals passing all quality controls were received. When only two sound source signals passed the quality control, computation of geographical position was attempted, but used only if the calculated position was consistent with the position before and after.

The computed positions were then compared with the GPS positions for all periods when both GPS fix and sound source echo were available. The differences between the GPS and the TOA-derived position ranged between 7 and 17 km, and this difference is considered here as a rough estimate of the absolute error in positioning. The seven trajectories are displayed in Fig. 3, with their smoothed trajectories obtained from the 24-hour low-pass filtered raw TOA.



## 2.3 Mooring and ship-based CTD data

In addition to the Lagrangian floats, in the present study we use the Eulerian time-series acquired at two mooring sites. One of the two moorings (M30.5W) was located at 76.09° S, 30.47° W (Fig. 1) from 22 January 2016 to 04 February 2018 (Ryan et al., 2020) at 445 m depth. It was deployed during the Polarstern cruise PS96 (Schröder, 2016) and recovered during the Polarstern cruise PS111 (Janout et al., 2019; Schröder, 2018). Temperature was recorded at 111, 97, 73, 48 and 33 m above bottom with additional salinity and pressure measurements at the shallowest and the deepest depths, and velocity at 18 m above bottom. The other mooring (P5) was located farther north at 75.39° S, 28.64° W at 437 m depth (Fig. 1). It was deployed on 09 February 2017 during the cruise of the float deployment and recovered on 09 March 2021 during the Polarstern cruise PS124 (Hellmer, 2020). Temperature was recorded at six different depths (7, 30, 55, 75, 104, 114 m above bottom) and velocity at three depths (6, 53, 102 m above bottom). Here we use the temperature and velocity measurements at 382 m and 384 m depth, respectively, which are the closest measurements to the float drift depth.

We also use the hydrography data from the ship-based CTD casts of the float deployment cruise to calculate the thickness of the ISW layer. We calculate the ISW layer thickness at each profile (both from the ship-based CTD casts and the float profiles) as the total thickness of all points where $\Theta < $ -1.9°C. We only include layers that are located in the bottom layer (with the lowermost measurement of $\Theta < $ -1.9°C at <50 dbar from the bottom), to exclude measurements of Winter Water that may also be colder than the surface freezing point but located higher up in the water column.

## 3 Results

### 3.1 Float trajectories

The profiling floats deployed on the Weddell Sea continental shelf drifted and profiled for three months to up to four years, providing hydrography and drift data over the eastern portion of sea ice covered continental shelf. With the use of three sound sources and RAFOS positions, we were able to recover the float locations with a frequency of six hours and of up to one year duration. The maximum distance from the sound sources for which good quality TOA were obtained was ~350 km (Fig. 2). This is a significantly shorter range than in open ocean experiments with sources emitting in the SOFAR channel (typically 800-1000 m depth; Rossby and Webb, 1970; Wong and Riser, 2011; Rossby et al., 1993), likely because of refraction, coupled with rough reflecting interfaces that attenuate the sound source signal. Our experiment deliberately avoided targeting flow under the ice shelf cavity to focus on the dynamics of the ISW and mWDW, and their interactions, on the continental shelf.

Two floats deployed in Feb. 2017 on the continental slope (P12682 and P12677; Fig. 3) followed the isobaths westward and then turned south onto the continental shelf at the opening to a small canyon cutting across the continental shelf between the Filchner Trough and Brünt Ice Shelf. We will here refer to this small canyon as Small Trough (Fig. 1). One of the two floats (P12682) moved southward through the Small Trough over the shallow eastern plateau until it reached the Filchner Trough at ~76.5° S in early May. On May 23, its trajectory then sharply veered from a southward direction to a northward flow along the lower eastern flank of the Filchner Trough. It continued profiling within the Filchner Trough, but without information on




its position (Fig. 4c). The other float (P12677) started off in a similar direction, but it stopped recording on May 20, when it

was still located within the Small Trough on its way towards the Filchner Trough.

The two floats deployed Feb. 2017 on the Filchner Trough Sill (P12686 and P12703, Fig. 3) were deployed only 6 km and 7 hours apart, but followed very distinct routes. One of the two (P12703) started off northward towards the continental slope, but turned southward on February 28 and continued above the eastern plateau towards the Filchner Trough. It reached ∼76.7°S on May 1, where it abruptly turned northward again at a similar location to where P12682 changed direction. In contrast, the

195 second float (P12686) started off on a southward track close to the upper eastern flank of the Filchner Trough but changed its direction abruptly at ∼75.5° S, where it drifted back northward along the lower eastern flank of the Filchner Trough. It continued westward along the Filchner Trough sill, leaving the Filchner Trough to follow isobaths over the continental slope.

All four of these floats (P12682, P12677, P12686 and P12703) had southward and relatively straight trajectories at different locations so long as they remained over the eastern shallow plateau, or upper eastern flank, above 400-500 m bathymetry depth.

As soon as the floats reached deeper bathymetry in the Filchner Trough, their direction changed and they ultimately drifted back northward.

To capture the flow of the ISW within the Filchner Trough, two additional floats were deployed directly inside the Filchner Trough, at ∼76.5° S (P12679 and P12684; Fig. 3). One of these two floats (P12679) could only be positioned for one month, during which it followed an eddying pattern close to its deployment position; however, it was able to profile the water column

for more than a year, which it spent entirely within the Filchner Trough as indicated by the sampled ISW layer (Fig. 5c,d). The second float (P12684) went northward toward 75.4° S following the eastern slope of the Filchner Trough before flowing back southward to 76.5° S. After several months of missing RAFOS positions, it reappeared at the western side of the Filchner Trough up to the Berkner Bank (39° W), providing Lagrangian positions more than one year after its deployment. Compared to the persistent southward trajectories on the eastern plateau, the trajectories of these two floats within the Filchner Trough

show a strong eddying pattern and recirculation within the Filchner Trough.

## 3.2 Modified Warm Deep Water and Ice Shelf Water regimes

The two groups of floats identified above appear to document two distinct regimes associated with two different water masses characterizing the eastern Weddell Sea continental shelf. While the first group of four floats (12677, 12682, 12686 and 12703) are associated with an eastern shallow plateau regime of mWDW, the second group of two floats (12679 and 12684) is associ-

215 ated with a Filchner Trough regime of ISW circulation.

To further highlight these two regimes, we investigate the temperature of the water column in the layers that host mWDW and ISW (Fig 6). mWDW is defined as the water-mass of temperature greater than -1.7°C in the layer of potential density $\sigma_0 = 27.75 \pm 0.05$ kg m$^{-3}$, and ISW, as the water-mass of temperature lower than the surface freezing point (∼-1.9°C) in the bottom ocean layer (Nicholls et al., 2009). Based on these definitions, we describe below (i) the circulation and timescale of

220 advection of mWDW on the eastern plateau; (ii) the interplay between mWDW and ISW on the eastern plateau; and (iii) the residence time of ISW in the Filchner Trough.



The four floats drifting southward on the eastern shallow plateau are associated with temperatures greater than -1.6°C (Fig 6a), suggesting that the region is flooded with southward-flowing mWDW. Based on the float trajectories, the southward mWDW transport is most efficient on the shallow eastern plateau, compared to a pathway closer to the Filchner Trough. Indeed, float P12686, which has the southward trajectory closest to the Filchner Trough, does not reach far south but gets quickly entrained into a northward flow that advects the float out of the Filchner Trough and then westward along the continental slope. In contrast, the two floats (P12682 and P12703) that were drifting farther east closer to the coastline, were transported much farther south. Their trajectories also indicate that at least two main pathways occurred in 2017 for the advection of mWDW from the continental slope onto the continental shelf (Fig. 1): the Small Trough (P12682) and the northeastern corner of the Filchner Trough (P12703). It takes the floats about two months to reach the interior Filchner Trough from the shelf break: 64 days through the Small Trough and 54 days along the eastern flank of the Filchner Trough from the shelf break at 74.6°S (the two floats' northernmost common latitude) to 76.5°S (the two floats' southernmost common latitude). The flow of mWDW through the Small Trough is further supported by the velocity and temperature time-series taken at the sound source mooring P5 positioned there (Fig 8). The float passed within 8 km of this mooring and the trajectory derived from the mooring velocities at the approximate float drift depth agrees well with the float trajectory. These mooring observations show a south-southwestward flow throughout the whole year with temperatures in the mWDW-range from February to June.

Interestingly, neither of the two floats went farther south than ∼76.7° S, where the eastern shallow plateau narrows. Instead, they became entrained in a northward flow. This turning point of the floats also coincides with the southernmost extent of the warm water in the climatology of the temperature at about 400 m depth shown in Fig. 1. We suggest that this region poses a bathymetric constraint on the southward flow: the narrowing shelf forces the flow off the relatively flat, shallow isobaths of the plateau and onto the eastern flank of the Filchner Trough dominated by a northward flow and colder water masses. At this point, their abrupt turn is accompanied by eddy motion suggestive of mixing with ISW along the eastern flank (Fig. 6a). One of the floats (P12682) crossed a clear front between the mWDW and the ISW in the beginning of June when entering the Filchner Trough (Fig. 4c), which coincides with the abrupt switch from a southward flow, within the mWDW layer, to a northward flow within the ISW layer (Fig. 4d).

The strong horizontal shear on the eastern side of the Filchner Trough, inferred from the opposing float directions over the eastern flank of the Filchner Trough, is likely associated with instabilities causing the mixing suggested by the float-derived hydrography. Unfortunately, the set of Lagrangian trajectories does not provide enough spatial and temporal sampling to address mixing or instability questions fully here. We nevertheless attempted to locate features within the trajectories resembling mesoscale eddies (Appendix A) and quantify their characteristics. We found a number of potential mesoscale features along the eastern flank of the Filchner Trough with radii ranging from 5 to 10 km, consistent with the local size of the first deformation radius (Lacasce and Groeskamp, 2020). While only suggestive, this is an indication that the region is unstable and prone to eddy activity hence turbulent mixing between mWDW and ISW. Another mesoscale eddy is visible over the Filchner Trough sill in the trajectory of float P12679 that escapes the trough westwards. The eddy aligns with a small depression in the bathymetry. This bathymetric feature might, as is typical of topographically induced curvature in flow, enhance eddy development, and impact ISW overflow and its interaction with the off-shelf WDW.



The temperature of the water lying at the bottom of the continental shelf indicates the presence of an ISW layer (Fig. 6b) that is observed at all profiles within the Filchner Trough, but also at the bottom of the eastern shallow plateau at around 76°S. The ISW layer reaches to a depth of about 250 m within the area sampled by the floats inside the Filchner Trough (Fig. 5).

While the ISW layer here is up to 1000 m thick at the deepest part of the trough close to the ice front, the ISW layer on the shallower shelf is much thinner (Fig. 7a). The float profiles inside the Filchner Trough also show that the ISW layer persists all year round, with only small variations in depth and temperature (Fig. 5). The ISW layer on the eastern plateau is sampled by float P12703 during a short period in early April (Fig. 4e), shortly before it passed the mooring M30.5W that provides information on the time variability of the ISW layer in this shallower region (Fig. 7b). The mooring temperature records show

that the presence of an ISW layer was seasonally variable: ISW was present at the bottom of the mooring site during austral summer (December to March; Fig. 7b) before the continental shelf got flooded by mWDW during autumn and winter (March to October). After the ISW has been replaced by mWDW, there were still several intrusions of ISW onto the continental shelf, during which the float passed the mooring site. The seasonal interplay between the ISW and the mWDW on the shallow eastern plateau has previously been described in more detail in Ryan et al. (2017). Our floats show that the mWDW is advected from

the continental slope through the Small Trough and along the eastern flank of the Filchner Trough within a timescale of about 2 months, similar to the seasonal variability timescale found at the eastern flank of the trough.

The two floats deployed on the eastern flank of the Filchner Trough (P12679 and 12684) drifted at the upper limit of the thick ISW layer (drift depth of about 250 dbar) that fills the Filchner Trough (Fig. 5). The two floats circulated inside the Filchner Trough for one and four years, respectively. Based on the existing positions, they circulated as far south as 76.64° S, and the

northernmost position is the last GPS position sent out by 12684 in January 2021 at 75.18° S. Despite the missing locations during large parts of the sampling period, the temperature measurements show the presence of a thick ISW throughout their whole measurement period. The depths of the profiles suggest that both floats moved from the shallow eastern flank of the Filchner Trough to the central or even western part of the trough, collecting profiles up to 984 m depth. For both floats, the temperatures within the ISW layer show a transition towards colder temperatures of the ISW during austral summer 2017/18

compared to the previous summer.

This result is in agreement with the change from the Berkner Mode towards the Ronne Mode described in Janout et al. (2021); Hattermann et al. (2021), where the ISW temperature changes based on the formation site of HSSW. It is striking that none of the two floats was advected northward out of the Filchner Trough in one year (for float P12679) or even four years (for float P12684), but instead recirculated within the Trough. This is indicated by both the position of the floats and, when

not available, by the hydrography and depth of the water-column sampled by the floats (Fig 5). The other floats however, that were advected northward along the eastern flank of the Filchner Trough, seem to have left the continental shelf eventually, as indicated by the trajectory of float P12686 that continued westward along the continental shelf west of Filchner Trough and by the hydrography of the two floats P12682 and P12703 that got entrained in the northward flow of ISW at about ∼76.7°S. The latter two floats started sampling a deep layer of warm water in austral summer 2017/2018 with a profile depth of more than

1600 m which can only be located over the deep ocean off the continental shelf. We conclude that the flow within the deeper



part of the Filchner Trough is recirculating within the trough, keeping the ISW layer trapped for several years, while the ISW closer to the eastern flank of the Filchner Trough is advected northward and across the sill relatively quickly (within one year).

## 4    Discussion

We deployed seven autonomous floats to document the circulation of two key water-masses on the southern Weddell Sea
continental shelf. The year-long Lagrangian trajectories within this sea ice covered region, as well as the up to four year-long full-depth CTD profiles along the float's trajectories, shed new light on pathways and timescales associated with mWDW inflow to the continental shelf, on the circulation of ISW within the Filchner Trough, and on the mixing between the two water masses.

The floats sampled the main inflow pathways of mWDW from the continental slope toward the continental shelf. Consistent
with previous work (Darelius et al., 2016; Årthun et al., 2012), our observations describe a southward flow of mWDW into the Filchner Trough across its sill at ~30° W. In addition, we demonstrate that mWDW is also advected southward through the Small Trough on the eastern shallower continental shelf, at ~28° W (Fig 1). This alternative passage onto the continental shelf was previously hypothesised but never demonstrated as a possible gateway for mWDW (e.g. Ryan et al., 2020). The flow of mWDW during austral summer through the Small Trough is further supported by current and temperature measurements taken
over one year by the mooring P5 located in the Small Trough. Also, consistent with this flow, Labrousse et al. (2021) described the presence of a large pool of mWDW in 2017 over the entire continental shelf east of the Filchner Trough, found as far east as 28°W, although their observations were slightly later in the year, in June and July. This collection of observations thus suggests that both the northeastern corner of the Filchner Trough and the Small Trough support a southward mWDW pathway in summer. Limitations on our observational period does not permit us to infer interannual variability, but we note that 2017,
the year in which we acquired most of our float observations, was described by Ryan et al. (2020) as a year of particularly warm inflow of mWDW. They proposed that the heaving of the Antarctic Slope Front was stronger in 2017 than in other years, allowing for warmer, saltier mWDW to intrude on the continental shelf and for the mWDW inflow to last longer.

All of the floats presented here that capture the southward flow of mWDW consistently depict the eventual entrainment of southward flowing mWDW into a northward flow along the eastern side of the Filchner Trough. Such a turning point in the
trajectories was also found in idealized modelling results from the Filchner Trough region by Daae et al. (2017), where in several experiments the southward flowing current along the eastern flank of the trough only reaches about 50 km into the trough before it gets drawn out of the trough again by the northward flow of ISW. This blocking of the mWDW inflow by the presence of ISW is consistent with the absence of mWDW close to FRIS that was earlier described from in situ observations (Darelius et al., 2016). Here, we provide a novel view on the southward extent of the mWDW pathway: mWDW entering
the continental shelf farther east from the Filchner Trough is able to penetrate a greater distance to the south toward the ice shelf. Hence, the Small Trough is of particular importance for the impact of warm water on the shelf. At the same time, our observations underline the importance of the ISW presence and its northward advection on blocking mWDW access to the





FRIS cavity. Any change in ISW production, which directly relates to the sea-ice driven production of HSSW (Hattermann et al., 2021; Janout et al., 2021), would therefore provide a feedback on changes in mWDW access to the FRIS cavity.

While our observations indicate that the northward ISW pathway on the eastern side of the Filchner Trough blocks mWDW access to the FRIS cavity, the two floats that sampled ISW in the deeper part of the trough showed convoluted trajectories and long residence time-scales of one year (P12679) and up to—possibly more than—four years (P12684), respectively. Such a long residence time was so far unknown and could only be revealed with under-ice float data. Although the reconstitution of trajectories was relatively poor for those two floats, their hydrography observations and irregular positioning (either through

RAFOS or GPS) indicate that ISW in the deeper part of the depression stays in the region. The outflow of ISW is therefore likely confined to specific pathways, topographically linked to the flanks of the Filchner Trough. The pathway of ISW outflow along the eastern flank of the Filchner Trough is clearly demonstrated by the four floats that first sampled the southward flow of mWDW before getting entrained in a northward flow of ISW as soon as they reached over deeper isobaths within the Filchner Trough. A northward pathway of ISW along the western flank is not observed by the floats, despite their presence on the

western flank. However, we cannot exclude an outflow of ISW on the western side of the trough, along which ISW has been observed to exit the FRIS cavity seasonally (Darelius and Sallée, 2018).

    Mesoscale activity has been proposed to be an important element of the southward transport of mWDW across the continental slope and onto the shelf (Stewart and Thompson, 2015; Stewart et al., 2018, 2019). So far, observational constraints imposed by difficult sea ice conditions have prevented direct observations of eddies; now, though, our float measurements

reveal eddy activity associated with the observed front between the southward flowing mWDW and northward flowing ISW. We acknowledge, however, that the low number of events sampled still precludes any statistically robust conclusion on eddy characteristics and fluxes in the region.

## 5    Conclusions

The first Lagrangian float observations were acquired over the Filchner-Ronne continental shelf from 2017 onward. This dataset

provides year-round observations, combining full depth profiles and high resolution (6-hourly) Lagrangian trajectories under sea-ice by acoustic positioning. Two mWDW inflow gateways are documented: the easternmost passage, the Small Trough, at ∼28°W, that was previously hypothesised to be a pathway but never before observed as a mWDW passage; the second passage is in the northeastern corner of the Filchner Trough, at ∼30°W. Both gateways are important for the seasonal mWDW flooding of the shallow continental shelf east of the Filchner Trough.

Our observations corroborate the importance of the exchanges between ISW and mWDW on the continental shelf, with the presence of a thick ISW layer within the Filchner Trough blocking mWDW access to the FRIS cavity, by consistently entraining and mixing mWDW within an efficient ISW northward flow over the eastern slope of the Filchner Trough. Within the Trough itself, the circulation of ISW appears to be more convoluted and is associated with significant recirculation, with a recirculation time-scale of up to at least four years, as indicated by one of our floats. The results from the deployment of

a set of autonomous floats on the ice-covered Filchner-Ronne sector of the Antarctic continental shelf is demonstrated here



to be a promising tool for the observation of year-round hydrography and circulation of the Antarctic continental shelves, in combination with other observational strategies.

*Author contributions.* JBS and LV conceived and designed the analysis. AM developed the algorithm for float positioning under the supervision of JBS and AL. NS and EP contributed to the visualisation of data on all the figures. AL and PL were instrumental in aquiring high

quality observations, especially around the deployment and programming of sound sources and floats. All authors contributed to the writing of the manuscript.

*Competing interests.* The authors have no financial or non-financial competing interests to report.

*Acknowledgements.* This study receives funding from the European Union's Horizon 2020 research and innovation program under grant agreement N°821001 (SO-CHIC), and from the European Research Council (ERC) under the European Union's Horizon 2020 research and

innovation program (grant agreement 637770). KS acknowledges support from NSF OPP-1643679 and NSF OCE-1658479. The authors would like to express their gratitude to the officers and crews of RV Polarstern (Cruise PS129) and the James Clark Ross (cruise JR16004) for their efficient assistance. Thanks to S. Ryan for the fruitful discussions. Thanks to T. Lebrun for his work on the floats data, M. Janout, M. Monsees and H. Le Goff for the mooring work.

*Data availability.* All Floats observations used in this paper are available here : https://doi.org/10.5281/zenodo.10353500



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





**Table 1.** List and characteristics of all floats deployed in 2017.

| Float | Deployment Location | Deployment time | Last CTD profile | Parking depth [m] | Number of profiles with RAFOS positions | Number of profiles |
|---|---|---|---|---|---|---|
| P12677 | -26.18822, -74.44278 | 10/02/2017 06:29 | 31/03/2020 | 400 | 49 | 220 |
| P12679 | -32.74232, -76.46532 | 18/02/2017 14:16 | 24/03/2018 | 250 | 11 | 82 |
| P12681 | -26.16269, -74.46104 | 10/02/2017 05:44 | 11/01/2021 | 400 | 31 | 137 |
| P12682 | -26.18320, -74.45266 | 10/02/2017 06:06 | 07/01/2021 | 400 | 49 | 220 |
| P12684 | -32.71802, -76.47313 | 18/02/2017 14:47 | 14/01/2021 | 250 | 38 | 127 |
| P12686 | -30.71351, -74.84898 | 12/02/2017 22:26 | 25/01/2018 | 400 | 29 | 90 |
| P12703 | -30.64634, -74.85276 | 12/02/2017 22:48 | 30/10/2020 | 400 | 75 | 192 |



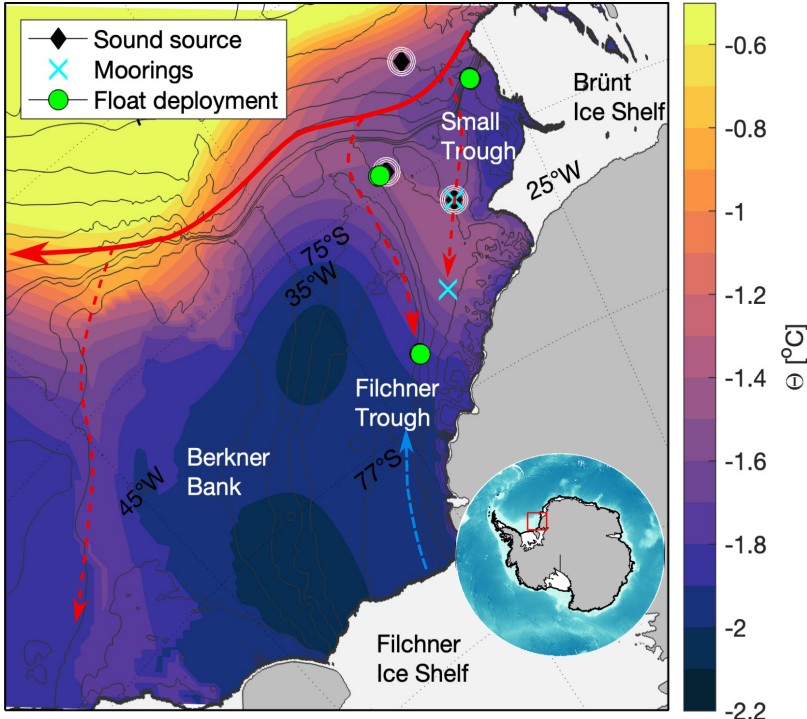

**Figure 1.** a) Map of the eastern Weddell Sea continental shelf with a schematic of the circulation. The arrows show the characteristic pathways of WDW (red), mWDW (dashed red) and ISW (blue). The colored background is the mean Conservative Temperature (Θ) between 360 and 420 m, from Jourdain et al. (2020). Bathymetric contours from IBSCO (Arndt et al., 2013) are drawn every 100 m above 1000 m depth and every 1000 m thence. Grounded ice is shown as gray areas and ice shelves as white areas. The black diamonds circled represent the position of the sound source moorings, the cyan crosses the position of the two moorings used in the study and the green dots the launching sites of the floats. The inset in the lower right shows the location within Antarctica.





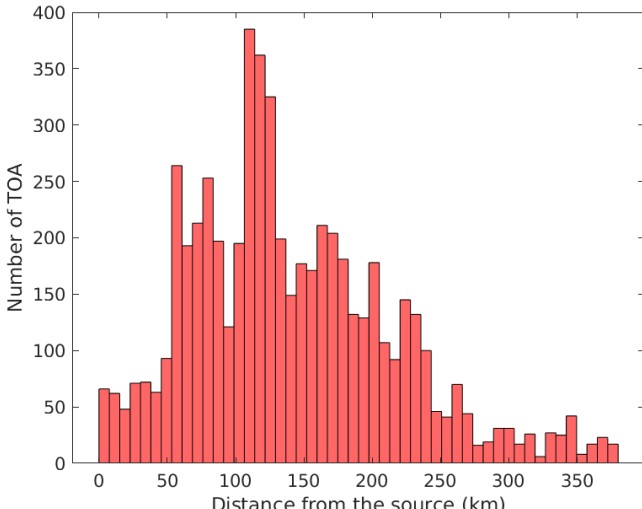

**Figure 2.** Histogram of the maximum distance between the floats and the sound sources. Distances of the floats to the sound source are used for all the positions recovered with two or more Time of Arrival (TOA) records.



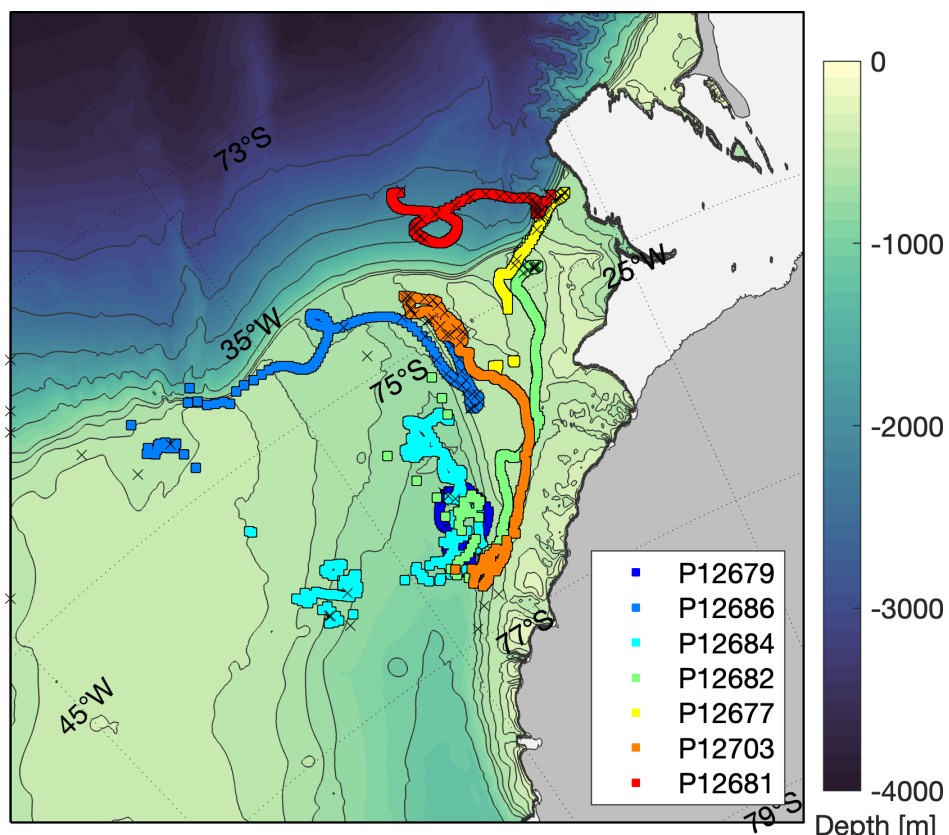

**Figure 3.** Drifting positions of the seven floats deployed in 2017. The smoothed positions computed from the RAFOS system are shown in color (squares; float 12681 is displayed in pink, 12677 in orange, 12682 in yellow, 12703 in red, 12679 in blue, 12684 in green and 12686 in turquoise). The crosses represent the GPS positions of the floats. The colored background represents the bathymetry, with contours as in Fig. 1.





**Figure 4.** (a,c,e,g) Hovmöller diagrams of Conservative Temperature for the floats that sampled mWDW on the shelf, Floats 12677, 12672, 12703 and 12686. The white contours represent the drift depth of the floats between the profiles and the black contours is the -1.9$^{o}$C isobath for the ISW. The triangles at the upper x-axis show the timing of the profiles, where those with existent RAFOS position are colored black. (b,d,f,h) shows the RAFOS trajectories of the floats on top of the bathymetry (as in Fig. 1), with the drift positions in gray and the profile positions in black. The colored circles are the existing positions closest to the 1st of each month, corresponding to the colored triangles at the lower x-axis in the Hovmöller diagrams. Although the CTD timeseries are up to four years, we only show the first year in which we have RAFOS positions and when the floats are still over the continental shelf.



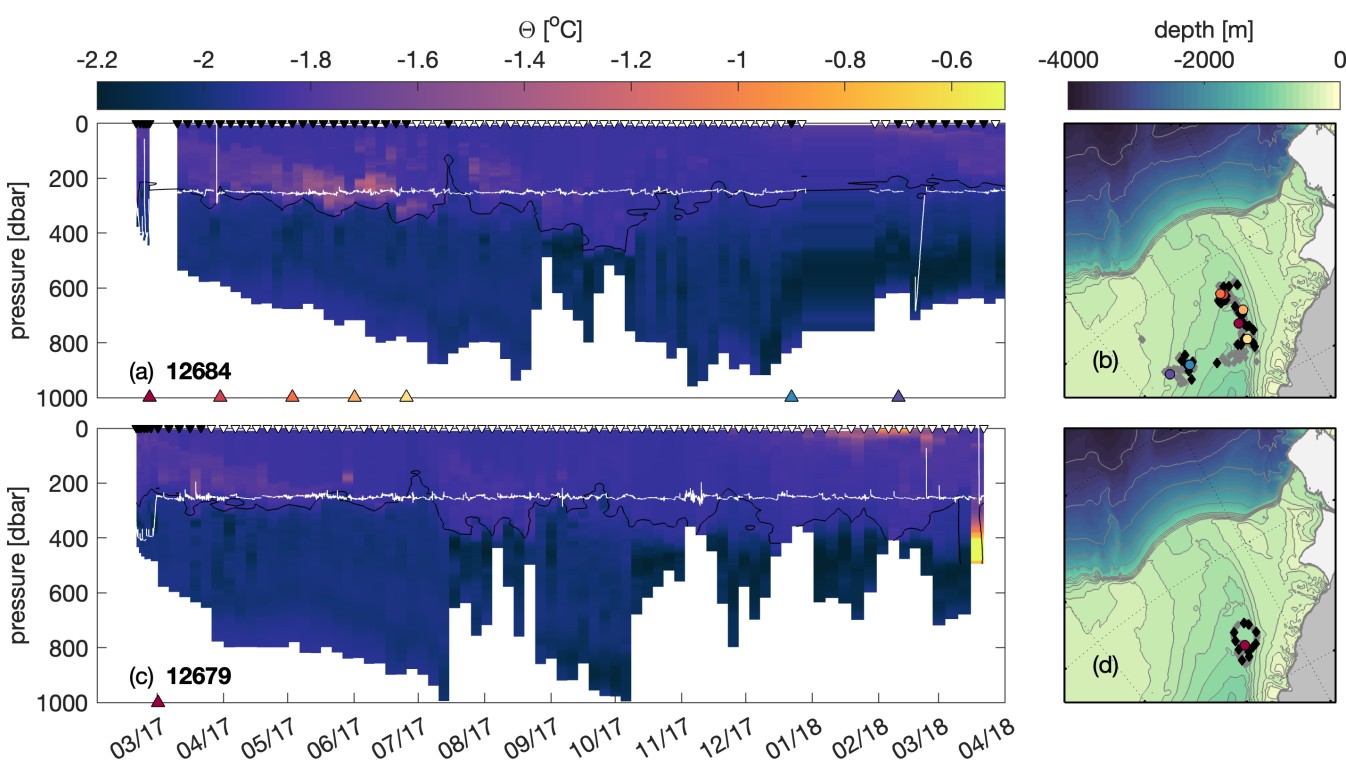

**Figure 5.** Same as Fig. 4, but for the floats that sampled ISW within the Filchner Trough, Floats 12684 (a,b) and 12689 (c,d).





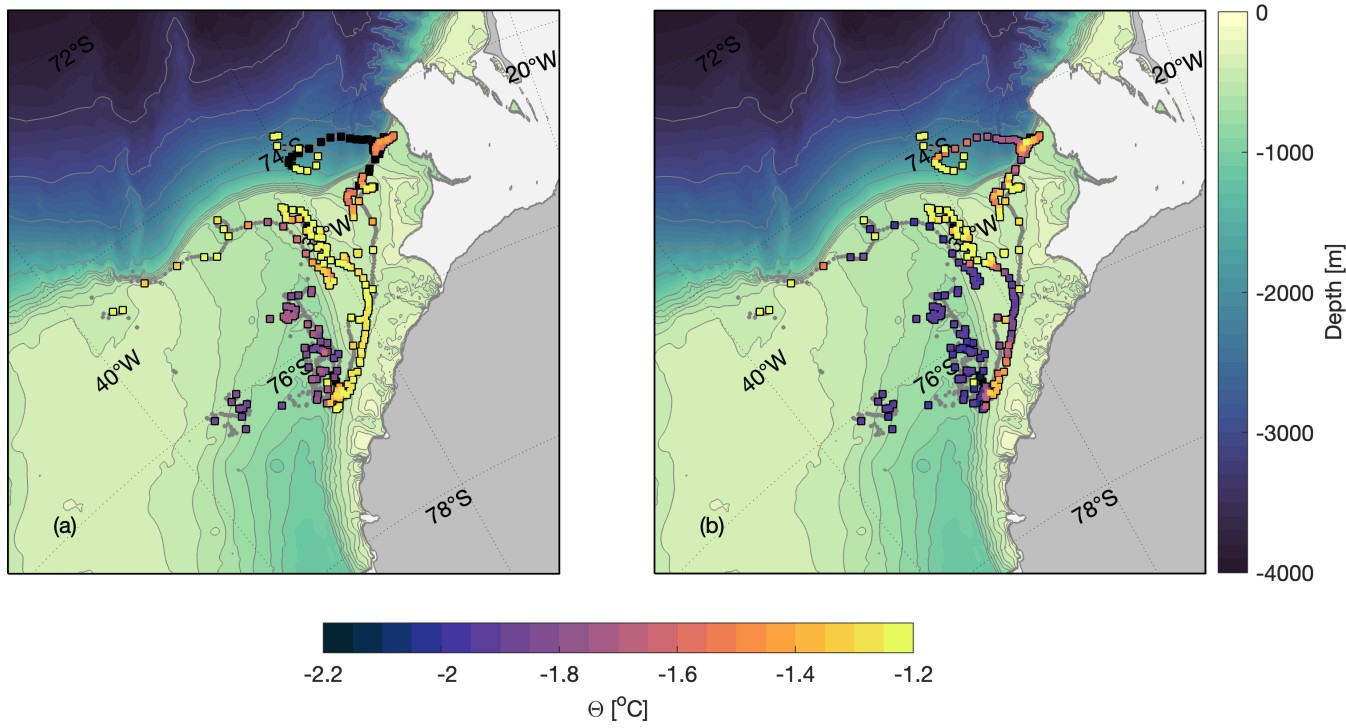

**Figure 6.** a) Conservative temperature (Θ; colored squares) of the float profiles at the 27.75 kg m³ ± 0.05 kg m³ density, showing the Θ in the mWDW density range. b) Bottom Θ of the float profiles (colored squares). The grey lines in a) and b) are the RAFOS drifting positions of the floats. The colored background represents the bathymetry with contours as in Fig. 1





**Figure 7.** a) ISW layer thickness from the float profiles (squares) and the ship-based CTD casts during the deployment cruise in 2017 (triangles) over the Weddell Sea continental shelf. The colored filling of the symbols show the thickness of the ISW layer (only if any ISW present), taken as the bottom water mass below the surface freezing point (-1.9°C). The colored background represents the bathymetry with contours as in Fig. 1). b) Hovmöller of the Conservative Temperature (Θ) of the mooring M30.5W (see its position in panel a)). The black contour represents the -1.9°C contour for ISW and the gray contours is the -1.7°C contour for mWDW. The red triangle shows the timing when the float P12703 passes closest to the mooring location. The black diamonds show the depths of the temperature sensors.



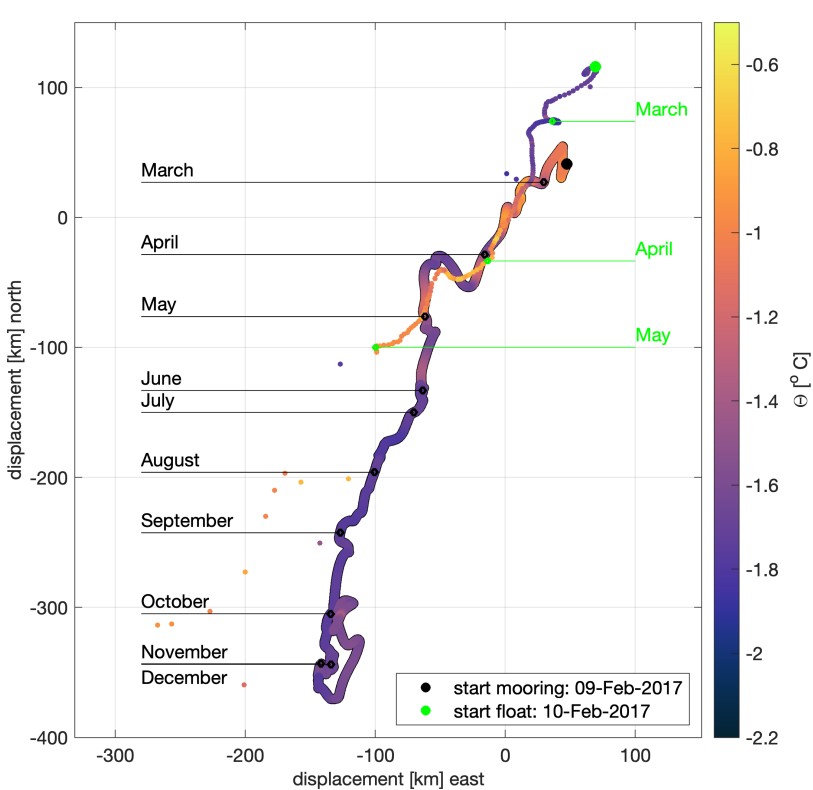

**Figure 8.** Progressive vector diagram of the current at 430 m depth measured at the sound source mooring P5 located in the Small Trough (thick line; see mooring location in Fig. 1) and of the trajectory of float P12682 that passes the mooring location (thin line). The displacements are centered around the time when the float passed the mooring for better comparison. The color shading denotes conservative temperature (Θ). The black (mooring) and green (float) circles along the trajectories mark the first day of each month according to the labels on the side in corresponding colors, and the first measurement date is marked with big circles according to the legend.



**Appendix A: Mesoscale Features**

The finely resolved Lagrangian trajectories of the floats allow us to attempt documenting mesoscale features that have influenced the float's drift. The float's movements are indeed characterized by convoluted trajectories that are reminiscent of trajectories influenced by a combination of larger-scale currents superimposed on smaller-scale variability. Here, we aim at documenting that smaller-scale variability. For that purpose, we use two different methods. First an ad-hoc method where we identify, by eye, all features in the trajectories that resemble what could be a mesoscale loop. A "loop" was defined as portion of the trajectory that cross itself within a small time difference. Second, we use a Bivariate Empirical Mode Decompositions (Bi-EMD, see below), to mathematically delineate low frequency paths and high frequency variability.

We identified, by eye, looping features, which we next refer to as "mesoscale loop": whether or not these can actually be called mesoscale will be further discussed below. Among the four trajectories of the floats that sampled long enough and that got positions for more than a month on the Weddell Sea continental shelf, we identify 11 features on four trajectories (P12682, P12703, P12686, P12684) that could be associated with "mesoscale loops" (Fig. 3). These are characterized by their representative diameter (largest distance between two points of the float trajectory inside the loop) and rotation speed (Table A1). Their diameter ranges from ∼2-20 km (median: 8.62 km), with a rotational speed ranging from ∼3-15 cm s$^{-1}$. We note that this ad-hoc method is associated with methodological bias: first, the loops can be largely distorted by the background flow; second, the float can be positioned at varying distance from the centre of the loop affecting the recovered characteristics.

For each trajectory where a "mesoscale loop" was identified by eye, we now mathematically decompose it into modes and show the highest mode as well as the residual trajectory. The used method, Bi-EMD, is a data driven method for the analysis of non-stationary data (Rilling et al., 2007). The method operates by decomposing a "bivariation", e.g., two-dimensional displacement, mathematically represented as a complex-valued signal, into a number of oscillatory modes. The version used in this study is the *cemdc2_fix* method issued from the EMD package from G. Rilling which is a fast implementation for bivariate/ complex EMD with a predefined number of iterations. The method does not work on all the 11 features, but provides an alternative objective method to describe the diameters of each located eddies, which are overall consistent with the diameter as computed with the ad-hoc method (Table A1).



| Floats | Diameter (km) | Pressure (dbar) | Time (days) | $\overline{V}_{eddy}$ (m/s) | $\overline{\Theta}$ (°C) | | Latitude (°S) | EMD diameter (km) |
|---|---|---|---|---|---|---|---|---|
| P12686 | 21.49 | 401 | 3.5 | 0.16 | -1.03 | C | 74.45 | 16.6 |
| P12686 | 3.300 | 400 | 3 | 0.035 | -1.65 | C | 75.55 | 2.73 |
| P12682 | 10.34 | 397 | 5 | 0.035 | -1.70 | C | 74.53 | 12.1 |
| P12682 | 3.061 | 398 | 2.75 | 0.022 | -1.73 | C | 74.88 | 3.92 |
| P12703 | 13.04 | 404 | 3.5 | 0.087 | -0.93 | AC | 75.06 | 8.05 |
| P12684 | 5.44 | 346 | 1.75 | 0.049 | -1.96 | AC | 76.14 | 3.18 |
| P12684 | 11.59 | 248 | 4.25 | 0.070 | -1.89 | AC | 75.52 | 5.69 |
| P12684 | 8.12 | 250 | 5 | 0.050 | -1.78 | AC | 75.52 | 8.49 |
| P12684 | 11.42 | 250 | 6.25 | 0.045 | -1.72 | C | 75.56 | 8.16 |
| P12684 | 8.62 | 250 | 7 | 0.042 | -1.73 | C | 75.61 | 9.61 |
| P12684 | 6.55 | 250 | 5 | 0.060 | -1.62 | AC | 75.90 | 4.83 |

**Table A1.** Description of the mesoscale structures selected by eye along the float's trajectory (see Fig **??**). For each mesoscale structure, its characteristics are provided: diameter computed from the ad-hoc method (km); pressure of the drift (dbar); time spent in the mesoscale structure (days); velocity of the displacement within the mesoscale structure (m/s); temperature at the pressure of drift within the mesoscale structure (°C); type of mesoscale structure: cyclonic (C) or anticyclonic (AC); latitude where the mesoscale structure was observed (°S); diameter computed from the EMD method (km).