# Peer review of "Subsurface floats in the Filchner Trough provide first direct under-ice tracks of the circulation on shelf"

_EGUsphere, 2023_

## Referee Comment (RC1)

**Review of Salleé et al. (2023)**

I find this paper to be an important contribution and enthusiastically recommend its publication. I do however recommend some minor revisions, and have three main comments for the authors. Firstly, I think the importance of this work is considerably under-sold, and think that it would be more impactful if the authors were more clear about what was novel. Secondly, I recommend moving the float processing to an appendix. Thirdly, I'm not at all convinced by the underdeveloped treatment of eddies and recommending cutting this part of the paper. Finally I have a number of minor presentational comments that should be easy to address.

**1 Clarifying the novel results**

It's important to be clear about what exactly is new. In the abstract, we are told "the mWDW flowing onto the continental shelf follows two pathways" and that these are the first under-ice Lagrangian measurements in the region. Later, we are told "there are two pathways for WDW to enter the continental shelf", citing earlier studies. It is essential to clarify how these facts are related. For example, if you could say "we provide the first in situ confirmation of two WDW pathways that have been indirectly inferred by earlier studies", that would add a lot of heft to the results. For this reason, when the two pathways are introduced beginning on line 53, it is important to give more details: are these known or directly measured? How were they inferred? What types of measurements were used? Etc. Otherwise, as it is presented, it looks like you could say "As everybody knows" in the abstract, which I'm sure is not the case.

Several other results encountered in the paper strike me as potentially novel, and if so, worth emphasizing: (i) first direct measurements of transit times for outflowing ISW and inflowing WDW; (ii) clear documentation of a sharp transition between southward and northward flow along the eastern flank of the Filchner Trough at 400–500 m (can you be more specific?); (iii) direct observations of a pool of an apparently stagnant pool of ISW, associated with high EKE region and long residence times, in the deeper part of the trough offshore of the transition isobath. All of these fact are mentioned, but they add up to a clearer picture of the behavior of the overall region than comes through because they are mentioned in isolation and are not adequately discussed within the context of what is already known. I would recommend to the authors to decide what the novel results are, and then make these crystal clear by itemizing them from the start, and returning to them in turn in dedicated paragraphs or perhaps even subsections where the direct and indirect evidence is all laid out.

A final factor I think the authors could emphasize further is the possibility of a regime transition if the ISW did not block the inflowing WDW. The hypothesis that the presence of ISW blocks WDW and therefore modulates the heat delivered to the ice sheet is enticingly suggesting by these results. Fleshing out this hypothesis with a description of "what-if" simulations or additional observations that could be carried out would be a powerful way to conclude the paper.

**2 Float positioning**

The gory details of float positioning should in my opinion be relegated to an appendix, as similar procedures have been followed in many previous studies. Although it is of course an essential part of the work, only a small percentage of readers would likely be interested. It would however be helpful to cite any papers (other than Rossby's original) that the authors are patterning their processing off of, e.g. Girton et al (2019), and also mention which if any aspects the processing are innovative or different. Finally the details of the loess filter should be mentioned because this is not a single filter but a family controlled by multiple parameters.

Out of curiosity, how did you deploy these floats? It seems relevant to mention whether and how they were deployed under ice.

**3   Don't talk about eddies**

The authors mention eddies and eddy generation in the title and abstract, but this is only discussed very briefly around lines 250–255 and again around line 340, together with Appendix A and Table 1. This is neither central to the authors' results nor well-developed. I would recommend to cut all of this. There are rigorous methods to exact eddies from Lagrangian data, but the bivariate EMD is not one of them. While this method does split a Lagrangian time series into pieces, the relationship of those pieces to the signals generated by eddies has never been examined, and might be nonsense. Furthermore, looking for isolated loops is a terrible way to identify eddies, despite the fact that it is often used. All the authors need to do to make their point is to look at the bandpassed or lowpassed velocity variance, and say that this is higher within the ISW-dominated deeper part of the Filchner Trough and that this higher variance, which will undoubtedly be isotropic, likely indicates enhanced eddy activity. This will be sufficient for the paper without over-reaching. A detailed survey of eddies would be deserving of its own paper.

**Minor comments**

1. Some minor typographic errors are indicated in the annotated document.
2. Generally speaking, I prefer to explicitly be introduced to figures in narrative form, e.g. "To investigate the ISW layer distribution and variability, Figure 7 shows the ISW thickness computed as discussed at the end of Section 2", rather than implicitly through reference as a part of the ongoing discussion. This way, the reader is clear about what each figure is intended to show.
3. "targeting the sources of modified Warm Deep Water (mWDW) inflow and Ice Shelf Water (ISW) circulation" is a little unclear; how about "at three locations of the eastern Weddell Sea continental shelf, targeting the inflowing modified Warm Deep Water (mWDW) as well as the outflowing Ice Shelf Water (ISW)"
4. line 5, would it not be better to describe this region simply as the continental shelf of the eastern Weddell Sea? I don't find " between the Brunt Ice Shelf and Filchner Ice Shelf" to be particularly helpful. The region is significant because is it the exchange region between the shelf and the deep flow of the continental slope, right?
5. line 10, use "via a small trough" here because the name " Small Trough" has yet to be introduced.
6. line 24, "formation of *the* Antarctic Bottom Water"
7. line 64, "However, the region has been suggested to be prone to a potential self-reinforcing change in the ocean circulation that could abruptly shift its cavity regime, turning it from a cold and dense cavity into a warm cavity with important global consequences for deep water formation and sea level rise." This is very interesting. I would propose the authors consider mentioning this fact much earlier, perhaps even in the abstract, and using it to underscore the importance and urgency of their study.
8. line 76 "a fixed point" should be "fixed points"
9. line 77, it is the not the water column that is vulnerable.
10. line 82, "employ" may be a better choice than "utilize"
11. line 84, What exactly is new is not clear from this sentence; it could be the use of these particular floats, or the use of any Lagrangian floats.
12. line 87, it is not clear whether you are referring to physical risk to the researchers or risk to the floats.
13. line 95, "complement" should be "complementing"
14. line 100 and following, minute symbols are consistently lacking on positions, e.g. 30°4' W. However, since the rest of the paper uses decimal degrees rather than degrees and arc-minutes, perhaps that convention should be followed here as well.
15. line 106, "sampled *a total* of"
16. line 130, "*with* frequency increasing linearly"
17. line 131 and elsewhere, "emission" should be "transmission"

18. line 132, "between *the* transmitted and expected"
19. line 148, since the notation $\Delta d$ is only use this single time, it's not necessary to introduce this notation.
20. line 172, I don't understand whether "with the lowermost measurement of $\Theta < -1.9°$C at $< 50$ dbar from the bottom" is describing the observed distribution of ISW or a condition applied to observations in other that they be used for the ISW thickness computation.
21. line 175, "for a time period of three months to four years" would be better.
22. line 180 "*acoustic* refraction" would help
23. line 183, "under the ice shelf cavity" should either be "under the ice shelf" or "within the cavity under the ice shelf."
24. line 228, please say roughly how many kilometers father south.
25. line 228, what about the word "entrances" rather than "pathways"?
26. line 232, "latitude) to reach" would be better.
27. line 232, it is not the flow that is supported. What about, "the notion that the Small Through provides a conduit for mWDW from the continental slope to the Filcher Trough is further supported..."
28. line 237, "neither of *these* two floats"
29. line 243, "*the floats'* abrupt turn"... "*eddying* motion", because you are describing the character of the flow and not necessarily talking about eddy structures.
30. line 246, I am not sure what "mixing suggested by the float-derived hydrography" means; I do not believe this has previously been discussed. It would be sufficient to say "associated with instabilities and the resulting mixing of water masses".
31. line 253, Regarding "Another mesoscale eddy is visible over the Filchner Trough sill in the trajectory of float P12679 that escapes the trough westwards", please point to the figure(s) where we can see this.
32. line 255, "This bathymetric feature might, as is typical of topographically induced curvature in flow, enhance eddy development, and impact ISW overflow and its interaction with the off-shelf WDW." This is pretty speculative; I would recommend moving it from Results to Discussion.
33. line 260 or thereabouts, you might refer back to the end of Section 2 where you discuss estimating ISW thickness; by the time I got here, I had forgotten how you calculated it.
34. line 271, why is it relevant to compare the advective and seasonal timescales?
35. line 282, it has been quite some pages since we last heard about HSSW; as this acronym is infrequently used, it would be clearer to write it out.
36. line 283, "*neither* of the two floats"
37. line 285, "that" should be "which" here.
38. line 303, "This alternative passage onto the continental shelf was previously hypothesised but never demonstrated as a possible gateway for mWDW", we should be informed of this much earlier, starting with the abstract.
39. line 304, see the second comment for line 232. I would start this off with something like "Other evidence supporting this notion of an inflow pathway through the Small Trough may be found in the literature."
40. line 321 "is of particular importance for the impact of warm water on the shelf" is a bit strong; how about "may be"?
41. line 345, for a paper of this length, I would recommend combining the conclusions and discussion into a single section.
42. Latitude and longitude should be indicated as such in the table.
43. In Fig. 1 and Fig. 3 it would be good to present an approximate distance scale. The map projection being used should also be noted.
44. Deployment locations should be marked in Fig. 3 so we know from this figure which way things are moving.

45. Same for the map insets in Fig. 4.
46. The meaning of the triangles in Fig. 4 is not clear to me from reading the caption. If the triangles are the same as the circles, the same symbol should be used.
47. Labelling the major water masses in Fig. 4 would be extremely helpful.
48. In Fig. 6, are the black symbols near the 74° S text meaning -2.2deg C? It seems very strange that the temperature should abruptly jump from -1.2deg C to -2.2deg C and back, as the colorbar would indicate. Or are these points actually off the color scale on the high end?
49. Again, if the red triangle and red circle are the same, the same symbol should be used.

[revised manuscript text omitted]

---

## Author Comment (AC1)

**Reviewer 1 : please find below our point-by-point response to reviewer #1. We copied reviewer comments in black. Our response is in orange. Citations from the manuscript are indicated in orange italicised text, and line numbers correspond to line numbers in the new version of the manuscript.**

Sallée et al present a fascinating study of trajectories and watermass transport implications from 7 RAFOS-enabled profiling floats on the eastern Weddell Sea continental shelf of Antarctica (Filchner Trough region). The floats operated in a mode similar to floats in the global Argo array but with a 5-day profiling interval (daily profiling in the summer), drifting at 250m or 400m between profiles and receiving RAFOS positions up to 4 times daily, with total float lifetime ranging from 1 to 5 years. Acquiring this dataset is in itself an impressive accomplishment.

We thank the reviewer for their careful reading of the manuscript and for providing such an enthusiastic comment on our work.

--- General Comments ---

The manuscript's presentation of the scientific context, the float experiment and results, and the discussion of implications is well done and informative. Figures are generally clear. In particular, the floats are simultaneously able to illustrate flow pathways and the evolution and variability of layer temperature and thickness along those pathways. Overall, not a lot of changes are needed before publication, although I do have a number of suggestions and questions which the authors may choose to address.

Thank you. We have considered all the comments made by the reviewer, as detailed below.

Although the discussion of pathways and transport mechanisms is interesting and informative, there has not been an attempt to make quantitative estimates of volume or heat transport or diffusive watermass transformations. Clearly any attempt to do this would be subject to some guesses at unmeasured quantities (such as flow width or duration), but even very rough values would be helpful for comparing these observations to numerical models or shipboard or moored measurements.

Although we very much agree with the reviewer that it would be great to have such estimates, we believe we are not in a position to be able to make them. The sparse nature of the float dataset presents significant challenges and limitations for making accurate estimates of the lateral transport of volumetric quantities (like heat). Despite our efforts, we found that deriving a reliable estimate based solely on these float datasets is exceedingly difficult and would be too speculative to be useful. Hence we postpone these questions to future studies that include data-modeling.

In one especially interesting and curious feature of Fig.8 (the comparison of a float's measured trajectory and temperature with the progressive-vector trajectory and temperature from a mooring), the temperature at the float made a sudden change immediately before

reaching the mooring. Is there any suggestion of how this might have happened (either through diffusive heating or non-Lagrangian movement of the float)? The change brought the float measurement up to the mooring's temperature just before the closest approach, but the warm temperature had been present at the mooring considerably earlier. Later, the warm patch seems to leave the mooring but the float continues to follow it.

We believe the main reason is due to the non-Lagrangian behaviour of the float. Before reaching the mooring the warm layer is just below 400 m, so that the float does not drift in the warm layer (see Figure 4). Due to the topography rise the warm layer reaches 400 m, roughly at the location of the mooring, so that the float drift starts to sample it from there, and follows it as it continues flowing southward. The warm layer stops being sampled by the mooring at this date because of intermittency of the inflow which is active at the shelf break only in Feb-Apr.

The authors make the intriguing suggestion that ISW could block the heat flux of mWDW toward the FRIS. Is this energetically possible? (Meaning, what is the source of energy maintaining the potential energy barrier of the dense water and accompanying geostrophic front?) Or is this really a statement about a different cause? For example, that the atmospheric cooling that produces the ISW (say close to the ice face) removes all heat before it can reach the cavity? Or simply that the amount of on-shelf mWDW transport and heat flux here are small relative to the West Antarctic shelf?

The idea of this "blocking" is based on layer depths modulated by geostrophic balance: that the ISW lies above the deeper isobaths and the mWDW over shallower isobaths that do not continue into the cavity. Across the shallower layer a front between these water masses exists, due to large-scale heat and freshwater fluxes, and the mWDW is not able to enter the cavity geostrophically.

--- Specific Comments ---

The bathymetry in Fig 1 is difficult to make out against the temperature shading. Would a light color work better? Also some bathymetric contour labels would be helpful. The bathymetry is presented in a more readable form in other panels, but there is no definitive label of any isobath and the colorbar is too finely graded to help. How about making one isobath (e.g. 300m or 400m) thicker than the others to provide at least one clear reference (along with the fixed 100m contour interval).

Accepted. We made the 500m isobath thicker for reference.

Also on the subject of bathymetry, I'm curious whether the floats drifting at 400m ever got stuck on the bottom during their drift phase. From counting the number of 100m-interval contours on the shelf east of Filchner Trough (e.g., in Fig.3), most of the shelf appears to be shallower than 300m. However, the depth-time plots in Fig.4 show water mostly between

400m and 500m on the shelf. Does the float-inferred bathymetry actually match the mapped contours? Or is the shallowest contour shown not the 100m one?

The floats drifted with the currents, which itself is steered by the bathymetry. So, the floats stayed above the seafloor, deeper than 400 m, even if sometimes they were very close to 400 m, as shown in Fig 4.

I suspect that a T-S diagram would help clarify some of the discussion of watermasses and layers. And coloring by or otherwise indicating variations in latitude or water depth might be a good way to illustrate the location, density range, and T-S characteristics of the ISW and mWDW watermasses and the front between the two.

We appreciate the suggestion of adding a T-S-diagram and added a second panel to Figure 3 with a T-S-diagram of all the profiles taken by the floats, color-coded by the water depth.

In addition, it would be useful to mark some candidate isopycnals (e.g., important ones for mWDW and/or ISW) on the depth-time plots (figs 4,5,7).

In the time-depth plots, we added the 27.75 isopycnal, as suggested in Figure 4 and 5. They now contain the contour of the ISW and mWDW, which are the water masses discussed in this paper. Regarding Figure 7, the focus of the figure is ISW so we prefer keeping only the -1.9°C isotherm, consistent with the definition of ISW used in panel a of the same figure.

Ending the Fig.4 trajectories on the shelf but continuing the timeseries off the shelf is confusing (and it is difficult to tell whether this is a choice that has been made deliberately or due to a lack of GPS data). Is there at least an approximate trajectory or direction of the floats that left the shelf (12682 and 12703) that could be indicated? They must have reached the surface at some point to send the data back.

We agree that it is unfortunate that not all CTD profiles can be geographically located. However, even if the location is not available for the whole extent of the CTD time series, we would still like to include the extended CTD time series to show, e.g., that floats 12684 and 12679 remained within the ISW layer at least throughout one year, which can be inferred from the observed water masses. It is true that the floats have surfaced at some point to transmit the CTD data; however, this has usually happened after the floats have drifted out of the study area, or beyond the single year shown here.

To avoid confusion, we have now drawn a vertical line in the time-depth plots where the last TOA position is available, and thus the trajectories in the map.

--- Minor Comments/Typos ---

l.9 typo: Pobservations

Accepted. Corrected.

l.202-210. The description of the southern float trajectories is interesting in it's comparison to the behavior of the eastern floats but should also re-iterate the fact that these floats parked at a different depth (250m) than the others (400m).

Accepted. Corrected.

Line 215: *"Compared to the persistent southward trajectories on the eastern plateau, the trajectories of these two floats that are parked at 200 m (shallower than the eastern plateau floats) within the Filchner Trough show a strong eddying pattern and recirculation within the Filchner Trough."*

Fig.3 caption lists different colors for the floats than the legend in the figure. Which is correct? (I'm guessing it's the legend.)

Corrected. We removed the color description from the caption.

Fig.4 mentions float 12672 but probably means 12682

Corrected.

"Hovmoller diagram" is usually not what a timeseries of vertical profiles is called (at least in oceanography). Not sure whether there's a definitive definition, though. "Depth vs. time" or "profile timeseries" seems more common for the types of plots shown in Figs 4, 5, and 7. Usually Hovmoller is reserved for horizontal distance vs. time (and in particular as a way to emphasize wave propagation that might be seen in an animation of 2D structures changing in time).

Accepted. We changed to "Profile timeseries"

Fig.6 shows temperature on the 27.75 isopycnal (mWDW). Can this isopycnal be added to Figs 4+5?

As described in our response above, we have now added the 27.75 isopycnal, as suggested, in Figure 4 and 5. They now contain the contour of the ISW and mWDW, which are the water masses discussed in this paper.

Fig.7 shows ISW layer thickness. Mention the definition of this layer as water colder than -1.9C in the caption.

The caption explicitly says: *"The colored filling of the symbols show the thickness of the ISW layer (only if any ISW present), taken as the bottom water mass below the surface freezing point (-1.9°C)."*

---

## Author Comment (AC2)

**Reviewer 2 : please find below our point-by-point response to reviewer #2. We copied reviewer comments in black. Our response is in orange. Citations from the manuscript are indicated in orange italicised text, and line numbers correspond to line numbers in the new version of the manuscript.**

I find this paper to be an important contribution and enthusiastically recommend its publication. I do however recommend some minor revisions, and have three main comments for the authors. Firstly, I think the importance of this work is considerably under-sold, and think that it would be more impactful if the authors were more clear about what was novel. Secondly, I recommend moving the float processing to an appendix. Thirdly, I'm not at all convinced by the underdeveloped treatment of eddies and recommending cutting this part of the paper. Finally I have a number of minor presentational comments that should be easy to address.

We thank the reviewer for their careful reading of the manuscript and for providing such an enthusiastic comment on our work. We have considered all the comments made by the reviewer, as detailed in our responses below.

**1 — Clarifying the novel results**

It's important to be clear about what exactly is new. In the abstract, we are told "the mWDW flowing onto the continental shelf follows two pathways" and that these are the first under-ice Lagrangian measurements in the region. Later, we are told "there are two pathways for WDW to enter the continental shelf", citing earlier studies. It is essential to clarify how these facts are related. For example, if you could say "we provide the first in situ confirmation of two WDW pathways that have been indirectly inferred by earlier studies", that would add a lot of heft to the results. For this reason, when the two pathways are introduced beginning on line 53, it is important to give more details: are these known or directly measured? How were they inferred? What types of measurements were used? Etc. Otherwise, as it is presented, it looks like you could say "As everybody knows" in the abstract, which I'm sure is not the case.

Accepted.

Abstract: "*We provide the first Lagrangian in situ confirmation that the mWDW flowing onto the continental shelf follows two pathways: the eastern flank of the Filchner Trough and via a Small Trough on the shallow shelf farther east.*"

Line 53: "*Conductivity-temperature-depth (CTD) sections and mooring time-series have been only sporadically obtained in the area. These observations have suggested that there are two main pathways for WDW to enter the continental shelf and subsequently flow southward toward the ice shelf as a yet cooler and fresher version of the WDW, referred to as modified WDW (mWDW): an eastern pathway, along the eastern flank of the Filchner Trough, and a western pathway, in the Central Trough west of the Berkner Bank (Fig. 1; Nicholls et al., 2009, 2008; Ryan et al., 2017). The western pathway is associated with the largest heat content (Nicholls et al., 2008), and about half of the heat at the shelf break that actually reaches the FRIS edge (Davis et al., 2022). CTD observations have also indirectly suggested that mWDW could enter the continental shelf by a third pathway flowing through the "Small Trough" (see Figure 1) directly east of the Filchner Trough, but this pathway has never been directly measured.*"

Several other results encountered in the paper strike me as potentially novel, and if so, worth emphasizing: (i) first direct measurements of transit times for outflowing ISW and inflowing WDW; (ii) clear documentation of a sharp transition between southward and northward flow along the eastern flank of the Filchner Trough at 400–500 m (can you be more specific?); (iii) direct observations of a pool of an apparently stagnant pool of ISW, associated with high EKE region and long residence times, in the deeper part of the trough offshore of the transition isobath. All of these fact are mentioned, but they add up to a clearer picture of the behavior of the overall region than comes through because they are mentioned in isolation and are not adequately discussed within the context of what is already known. I would recommend to the authors to decide what the novel results are, and then make these crystal clear by itemizing them from the start, and returning to them in turn in dedicated paragraphs or perhaps even subsections where the direct and indirect evidence is all laid out.

Accepted. We clarified that aspect in the last paragraph of the introduction.

Line 94: *"Our observations on the eastern Weddell Sea shelf allow us to describe the circulation in the Filchner Trough region and the water mass pathways based on the float trajectories and the hydrography profiles, some of which are taken in previously unexplored areas that are typically covered with sea ice. We are able to examine the Lagrangian pathways of the mWDW toward the ice shelf, its interaction with the ISW, and the residence time of the ISW within the Filchner Trough. Selected mooring observations and ship-based measurements from the same time period complementing the float data are presented. This combined dataset allow us to present the first direct measurements of pathway and transit times for outflowing ISW and inflowing mWDW. In particular, we demonstrate the existence of a mWDW pathway through a small trough which was previously suggested, but was never directly measured. We also present the first direct observations of a pool of ISW with very long residence times of several years in the deeper part of the Filchner Trough."*

A final factor I think the authors could emphasize further is the possibility of a regime transition if the ISW did not block the inflowing WDW. The hypothesis that the presence of ISW blocks WDW and therefore modulates the heat delivered to the ice sheet is enticingly suggesting by these results. Fleshing out this hypothesis with a description of "what-if" simulations or additional observations that could be carried out would be a powerful way to conclude the paper.

**2 — Float positioning**

The gory details of float positioning should in my opinion be relegated to an appendix, as similar procedures have been followed in many previous studies. Although it is of course an essential part of the work, only a small percentage of readers would likely be interested. It would however be helpful to cite any papers (other than Rossby's original) that the authors are patterning their processing off of, e.g. Girton et al (2019), and also mention which if any aspects the processing are innovative or different. Finally the details of the loess filter should be mentioned because this is not a single filter but a family controlled by multiple parameters.

Given the importance of this procedure for the paper, we prefer keeping this section in the main core of the paper. However, we can reconsider this decision if the editor thinks the section should sit in Appendix. The second reason we want to keep in the core of the paper, is that we found that these kinds of procedures are poorly documented in past

papers, often relegated to hard-to-obtain reports. Hence we wish to provide details on our processing, as well as for clarity of purpose. We provided details for the loess method, as requested.

Out of curiosity, how did you deploy these floats? It seems relevant to mention whether and how they were deployed under ice.

These floats were deployed in summer in open waters, before the seasonal advance of sea-ice. We have clarified that point:

Line 108: "*The floats were deployed in open waters, before the seasonal advance of sea-ice.*"

**3 — Don't talk about eddies**

The authors mention eddies and eddy generation in the title and abstract, but this is only discussed very briefly around lines 250–255 and again around line 340, together with Appendix A and Table 1. This is neither central to the authors' results nor well-developed. I would recommend to cut all of this. There are rigorous methods to exact eddies from Lagrangian data, but the bivariate EMD is not one of them. While this method does split a Lagrangian time series into pieces, the relationship of those pieces to the signals generated by eddies has never been examined, and might be nonsense. Furthermore, looking for isolated loops is a terrible way to identify eddies, despite the fact that it is often used. All the authors need to do to make their point is to look at the bandpassed or lowpassed velocity variance, and say that this is higher within the ISW-dominated deeper part of the Filchner Trough and that this higher variance, which will undoubtedly be isotropic, likely indicates enhanced eddy activity. This will be sufficient for the paper without over-reaching. A detailed survey of eddies would be deserving of its own paper.

Accepted:

- We removed the Appendix entirely.
- We removed the description of eddies in the result section; only left the few descriptions of eddying motion in the trajectories.
- We changed the associated paragraph in the discussion to something much more speculative:

Line 255: "The strong horizontal shear on the eastern side of the Filchner Trough, inferred from the opposing float directions over the eastern flank of the Filchner Trough, is likely associated with instabilities and the resulting mixing of water masses. Unfortunately, the set of Lagrangian trajectories does not provide enough spatial and temporal sampling to address mixing or instability questions fully here. The trajectories of the floats show however some eddying motion at along the eastern flank of the Filchner Trough, which might be an indication of instabilities and eddy activity, hence turbulent mixing between mWDW and ISW."

**Minor comments**

1. Some minor typographic errors are indicated in the annotated document.

We thank the reviewer for the detailed reading and the long list of suggestions, which have almost all been accepted. This greatly improves the quality of the manuscript. We are very grateful for the effort of the reviewer for this.

2. Generally speaking, I prefer to explicitly be introduced to figures in narrative form, e.g. "To

investigate the ISW layer distribution and variability, Figure 7 shows the ISW thickness computed as discussed at the end of Section 2", rather than implicitly through reference as a part of the ongoing discussion. This way, the reader is clear about what each figure is intended to show.

Noted.

3. "targeting the sources of modified Warm Deep Water (mWDW) inflow and Ice Shelf Water (ISW) circulation" is a little unclear; how about "at three locations of the eastern Weddell Sea continental shelf, targeting the inflowing modified Warm Deep Water (mWDW) as well as the outflowing Ice Shelf Water (ISW)"

Accepted.

4. line 5, would it not be better to describe this region simply as the continental shelf of the eastern Weddell Sea? I don't find " between the Brunt Ice Shelf and Filchner Ice Shelf" to be particularly helpful. The region is significant because is it the exchange region between the shelf and the deep flow of the continental slope, right?

Accepted.

5. line 10, use "via a small trough" here because the name " Small Trough" has yet to be introduced.

Accepted.

6. line 24, "formation of *the* Antarctic Bottom Water"

Accepted.

7. line 64, "However, the region has been suggested to be prone to a potential self-reinforcing change in the ocean circulation that could abruptly shift its cavity regime, turning it from a cold and dense cavity into a warm cavity with important global consequences for deep water formation and sea level rise." This is very interesting. I would propose the authors consider mentioning this fact much earlier, perhaps even in the abstract, and using it to underscore the importance and urgency of their study.

We are reluctant to place that part in the abstract because of the word limit constraint.

8. line 76 "a fixed point" should be "fixed points"

Accepted.

9. line 77, it is the not the water column that is vulnerable.

Accepted. "*leaves **instruments** in the upper part of the water column vulnerable*"

10. line 82, "employ" may be a better choice than "utilize"

Accepted.

11. line 84, What exactly is new is not clear from this sentence; it could be the use of these particular floats, or the use of any Lagrangian floats.

Accepted. "*We present and use, for the first time in this region, observations from Lagrangian profiling floats.*"

12. line 87, it is not clear whether you are referring to physical risk to the researchers or risk to the floats.

Accepted. "*make float deployments very risky for the instruments*"

13. line 95, "complement" should be "complementing"

Accepted.

14. line 100 and following, minute symbols are consistently lacking on positions, e.g. 30°4' W. However, since the rest of the paper uses decimal degrees rather than degrees and arc-minutes, perhaps that convention should be followed here as well.

Accepted.

15. line 106, "sampled *a total* of"

Accepted.

16. line 130, "*with* frequency increasing linearly"

Accepted.

17. line 131 and elsewhere, "emission" should be "transmission"

Accepted.

18. line 132, "between *the* transmitted and expected"

Accepted.

19. line 148, since the notation $\Delta d$ is only use this single time, it's not necessary to introduce this notation.

Accepted.

20. line 172, I don't understand whether "with the lowermost measurement of $\Theta < -1.9°C$ at $< 50$ dbar from the bottom" is describing the observed distribution of ISW or a condition applied to observations in other that they be used for the ISW thickness computation.

We added "i.e." which should clarify: "*We only include layers that are located in the bottom layer (i.e. with the lowermost measurement of $\Theta < -1.9°C$ at $< 50$ dbar from the bottom)*"

21. line 175, "for a time period of three months to four years" would be better.

Accepted.

22. line 180 "acoustic refraction" would help

Accepted.

23. line 183, "under the ice shelf cavity" should either be "under the ice shelf" or "within the cavity under the ice shelf."

Accepted. Changed to "*under the ice shelf*"

24. line 228, please say roughly how many kilometers father south.

Accepted.

25. line 228, what about the word "entrances" rather than "pathways"?

We prefer sticking with the word pathway, but if the reviewer has a very strong opinion about it, we are open to reconsider.

26. line 232, "latitude) to reach" would be better.

Accepted.

27. line 232, it is not the flow that is supported. What about, "the notion that the Small Through provides a conduit for mWDW from the continental slope to the Filcher Trough is further supported..."

Accepted.

28. line 237, "neither of these two floats"

Accepted.

29. line 243, "the floats' abrupt turn"... "eddying motion", because you are describing the character of the flow and not necessarily talking about eddy structures.

Accepted.

30. line 246, I am not sure what "mixing suggested by the float-derived hydrography" means; I do not believe this has previously been discussed. It would be sufficient to say "associated with instabilities and the resulting mixing of water masses".

Accepted.

31. line 253, Regarding "Another mesoscale eddy is visible over the Filchner Trough sill in the trajectory of float P12679 that escapes the trough westwards", please point to the figure(s) where we can see this.

Accepted.

32. line 255, "This bathymetric feature might, as is typical of topographically induced curvature in flow, enhance eddy development, and impact ISW overflow and its interaction with the off-shelf WDW." This is pretty speculative; I would recommend moving it from Results to Discussion.

This part of the text has been removed from the revised manuscript.

33. line 260 or thereabouts, you might refer back to the end of Section 2 where you discuss estimating ISW thickness; by the time I got here, I had forgotten how you calculated it.

Accepted.

34. line 271, why is it relevant to compare the advective and seasonal timescales?

Agreed that is not so relevant and might be confusing. We removed the end of the sentence comparing with seasonal timescale.

35. line 282, it has been quite some pages since we last heard about HSSW; as this acronym is infrequently used, it would be clearer to write it out.

Accepted.

36. line 283, "neither of the two floats"

Accepted.

37. line 285, "that" should be "which" here.

Accepted.

38. line 303, "This alternative passage onto the continental shelf was previously hypothesised but never demonstrated as a possible gateway for mWDW", we should be informed of this much earlier, starting with the abstract.

Accepted. See our response to the reviewer's comment point "1 — Clarifying the novel results", above.

39. line 304, see the second comment for line 232. I would start this off with something like "Other evidence supporting this notion of an inflow pathway through the Small Trough may be found in the literature."

Accepted. Line 308: "Other evidence supporting this notion of an inflow pathway through the Small Trough is provided by current and temperature measurements taken over one year by the mooring P5 located in the Small Trough."

40. line 321 "is of particular importance for the impact of warm water on the shelf" is a bit strong; how about "may be"?

Accepted.

41. line 345, for a paper of this length, I would recommend combining the conclusions and discussion into a single section.

We prefer keeping two separate sections unless the reviewer thinks this is a major point that needs to be addressed.

42. Latitude and longitude should be indicated as such in the table.

Accepted.

43. In Fig. 1 and Fig. 3 it would be good to present an approximate distance scale. The map projection being used should also be noted.

Accepted. The approximate distance per longitude in this area in the legend of Figure 1, as well as information about the projection. "The lambert projection is used. In this area, 1° longitude corresponds to about 30~km. " Since we use the same projection in Figure 3, we do not repeat this information.

44. Deployment locations should be marked in Fig. 3 so we know from this figure which way things are moving.

We marked the deployment locations in Fig. 3, as suggested.

45. Same for the map insets in Fig. 4.

We marked the deployment locations in Fig. 4, as suggested.

46. The meaning of the triangles in Fig. 4 is not clear to me from reading the caption. If the triangles are the same as the circles, the same symbol should be used.

We changed the colored circles on the map to triangles to be consistent with the triangles in the left panels.

47. Labelling the major water masses in Fig. 4 would be extremely helpful.

In the time-depth plots, we added the 27.75 isopycnal. They now contain the contour of the ISW and mWDW, which are the water masses discussed in this paper.

48. In Fig. 6, are the black symbols near the 74◦ S text meaning -2.2deg C? It seems very strange that the temperature should abruptly jump from -1.2deg C to -2.2deg C and back, as the colorbar would indicate. Or are these points actually off the color scale on the high end?

The black symbols where not associated to the colormap, but referred to the absence of the watermass. We removed those symbols in the new version of the figure to avoid confusion with the colorbar.

49. Again, if the red triangle and red circle are the same, the same symbol should be used.

We changed the colored circles on the map to triangles to be consistent with the triangles in the left panels.

---

## Author Response (AR2)

**Reviewer 1 : please find below our point-by-point response to reviewer #2. We copied reviewer comments in black. Our response is in orange. Citations from the manuscript are indicated in orange italicised text, and line numbers correspond to line numbers in the new version of the manuscript.**

I'm quite satisfied with the author's reponses to my review. I'm happy to recommend publication after addressing the minor typographic and clarifying issues listed below. The figures in particular are beautiful. I also really appreciate that the authors clearly separate their description of the observations from their interpretations of the meaning of those observations. Overall I think this is tightly argued and well presented piece of work on an important region.

We thank the reviewer for their positive comment on our revision. We have considered all the minor comments made by the reviewer, as detailed in our responses below.

10, omit "via"
Accepted

54, "time series"
Accepted

55, "flow onto the continental shelf and subsequently southward" would be better
Accepted

59, I think that the authors are saying that the western pathway is estimated to contribute about 1/2 of the heat that is delivered to the FRIS; in this case, the "at the shelf break" could be omitted for clarity, or some other slight rewording
Accepted - "at the shelf break" dropped.

59, please provide a reference for the asserting beginning "CTD observations..."
Accepted. Janout et al., 2021 added.

67, I believe what is meant here is "Thus far, mWDW has only been observed at the ice fron thear 79 S once, in 2013". It's a little unclear what the "only" is intended to mean.
Reworded: Thus far, mWDW has been observed at the ice front near 79°S only once in 2013

75, "sea ice"
Accepted

77, "water masses"
Accepted

80, Argo floats would not be suitable for a shelf study because of their typically deep profile depth, so it would be better to simply say "Autonomous instruments"
Rejected. Argo floats have been used in continental shelf studies, including the present study.

89, "infer" or "triangulate" would be better than "provide"
Accepted.

92, I think what the authors are meaning to say is that the approach taken in the present study is similar to that taken in a recent study in a different region, but that is not coming through in the current wording.
Reworded.

100, since you've aleady introduced the term Small Trough, perhaps use it here?
Accepted.

101, "but never directly measured" is better
Accepted.

109, "sea ice"
Accepted.

136, "transmitted a frequency-modulated signal" (no with) is better
Accepted.

140 and afterward, since TOA is used to refer to multiple times of arrive, I believe it should be written as TOAs is almost all cases (line 148 and 149 are exceptions)
Accepted.

150, "unrealistic records were removed", "time series"
Accepted.

151, please specify the bandwidth of the LOESS filter for reproducibility
It is already mentioned in the sentence: "removing frequencies higher than once per day"

157, "postition was estimated" or "inferred", not "computed"
Accepted.

164, it is not clear if "smoothed trajectories obtained from ... " is referring to the process just decribed. If it would be better to just say it like that. "raw TOA" is confusing because it sounds like you are now referring to a different processing
Accepted. We removed the part of the sentence ", with their smoothed trajectories obtained from the 24-hour low-pass filtered raw TOAs" to avoid confusion.

165, "time series"
Accepted.

175, "hydrographic"
Accepted.

177, Can you reword this without saying layers twice?
Accepted.

183, "sea-ice-covered" I believe
Accepted.

184, six hours is not the frequency; it is the sampling interval.
Accepted.

186, "open ocean"; "transmitting" is better than "emitting"
Accepted.

193, "the Small Trough"
Accepted.

217, "and a recirculation"
Accepted.

222, "regime of recirculation mWDW on the eastern shallow continental shelf" is better
Accepted.

225 & 226, "water mass"
Accepted.

232, "appears to be more efficient... compared to" is better
Accepted.

242, "time series"
Accepted.

245, "mWDW range"
Accepted.

248, "that this region is the site of" or "that in this region there is" are better, since it is not the region that poses.
Accepted.

250, "that is dominated by"
Accepted.

256, "over the eastern flank of the Filcher trough" can be omitted.
Accepted.

258, ", however, "
Accepted.

259, ", hence turbulence mixing, "
Rejected, we prefer turbulent mixing.

275, "with a timescale" is probably what is meant hear, as "within" would mean two months or less
Accepted.

278, "available" is better than "existing"
Accepted.

286, semicolon should be "and". "in which" would be better than "where" as it would clarify that you are explaining the meaning of the mode change.
Accepted.

292, "were" is better than "got"
Accepted.

299, "water masses"
Accepted.

300, "sea-ice-covered"
Accepted.

308, "directly observed" is better than "demonstrated"
Accepted.

314, "Limitations... do not"
Accepted.

323, "out of the trough" can be omitted
Rejected. We prefer being clear here.

329, perhaps emphasize at the end of this sentence that there would then be a associated impact on the heat fluxes to the base of the ice shelf.

Accepted.

332, "recirculating" is better than "convoluted"; the latter is more often used to denote logically muddled than physically wiggly
Accepted.

334, It is not clear what "irregular positioning" means since "positioning can mean the act of inferring a position"; perhaps "available positions" is what is meant.
Accepted.

345, ", however,". "motion along"
Accepted.

346, "hence turbulent mixing, "
Accepted.

349, "first Lagrangian observations over the Filchner–Ronne ice shlef were acquired" is I think what is meant.
Accepted.

351, "sea ice". The phase after the colon is awkward and would be better phrases as "an eastern passage..., and a western passage..."
Accepted.

359, "time scale". "up to four years" is all that can be said based on the data, not "up to at least".
Accepted.

373, "float data" not "floats data"
Accepted.

375, "Float" is not capitalized.
Accepted.

388, journal name capiltalization
390, missing journal name
394, journal name capiltalization
403, missing journal name
433, article title capitalization
448, Southern Ocean
466, journal name capiltalization
477, missing journal name, tech report details, or link
486, article title capitalization

497, missing journal name

Thank you. The references have been checked.

Table 1, Please indicate whether the time given is UTC, local time, etc.

Accepted.

Figure 1, "thence" needs to be unpacked into more words to explain the deeper contours. "Lambert" should be capitalized.

Accepted. We clarified thence.

Figure 2, what does "maximum" mean here? I believe you mean that for all position fixes associated two or more TOA records, the distance from the float to the farthest sound source is reported, but that's coming through at the moment.

Accepted. We clarified by removing maximum.

Figure 3, a minor point, the cross are not very visible; small black dots or filled circles might be better. No hyphen after "S" in "T-S diagram". "estimated as the maximum profile depth" would be better than "inferred from the profile depth." Is the colorbar axis in the right panel the same as that used in the left panel? I'm seeing a lot more dark blue on the right then I would expect to see based on the left, suggesting perhaps the colorbar axes not the same. In this and future maps, heavy gray lines are float tracks while light gray lines are isobaths corresponding to changes in the shading color. You might suppress the latter as they are not really needed; otherwise, in this and following maps, please clarify that it is the heavy gray lines that are float tracks.

Accepted some of the minor comments. Note that the heavy gray line is the isobath contour as described in the caption "with contours as in Fig. 1".

Figure 4, the meaning of "for" in "for the ISW" and "for the mWDW" is not clear. Something like, e.g. "marking a typical value for" would be better. Second to last sentence, I believe "colored circles" should be "colored triangles".

Accepted. We clarified : The cyan contours are the -1.9°C isotherm, corresponding to the definition of ISW, and the red contours are the 27.75kg m-3 isopycnal, corresponding to the definition of mWDW.

Figure 8, in the last sentence, the construction "black (mooring) and green (float) circles is confusing; it would be better to first tell us about the black circles and then about the green circles.

Accepted. We reworded to avoid confusion.